# Carcass and Pork Quality and Gut Environment of Pigs Fed a Diet Supplemented with the Bokashi Probiotic

**DOI:** 10.3390/ani11123590

**Published:** 2021-12-18

**Authors:** Artur Rybarczyk, Elżbieta Bogusławska-Wąs, Bogumiła Pilarczyk

**Affiliations:** 1Department of Animal Nutrition and Feed Science, Wrocław University of Environmental and Life Science, Chełmońskiego 38C, 51-630 Wrocław, Poland; 2Department of Applied Microbiology and Human Nutrition Physiology, West Pomeranian University of Technology, Papieża Pawła VI 3, 71-459 Szczecin, Poland; elzbieta.boguslawska-was@zut.edu.pl; 3Department of Animal Reproduction Biotechnology and Environmental Hygiene, Faculty of Biotechnology and Animal Husbandry, West Pomeranian University of Technology, Janickiego 29, 71-270 Szczecin, Poland; bogumila.pilarczyk@zut.edu.pl

**Keywords:** pigs, probiotics, Bokashi, slaughter value, meat quality, mineral content, microbiota

## Abstract

**Simple Summary:**

The use of an EM^®^Bokashi probiotic preparation containing specific *Lactobacillus* and yeasts strains as a feed additive resulted in the improved slaughter value, content of macroelements (Mg, Ca, Na) and chromatic color traits (b*, C*) of meat, but diminished the technological quality (pH, drip loss, TY, shear force) of pork. It additionally resulted in a significant increase in lactic acid bacteria (LAB) and yeast counts and a decrease in the population numbers of *Clostridium* and *Enterobacteriaceae* in gut microbiota.

**Abstract:**

The aim of this study was to determine the effect of probiotics on gut microbiota, on carcass and meat quality and on mineral contents in the *longissimus lumborum* (LL) muscle in pigs. The research was carried out with 120 hybrid pigs deriving from Naïma sows and P-76 boars. Pigs from the experimental group received the EM^®^Bokashi probiotic (Greendland Technologia EM®, Janowiec n/Wisłą, Poland) in their feed (containing *Saccharomyces cerevisiae*, *Lactobacillus casei* and *Lactobacillus plantarum*). The study showed that EM^®^Bokashi probiotic supplementation resulted in a significantly higher count of lactic acid bacteria (LAB) and yeasts in the feed, a lower number of *Clostridium* in the mucosa and colorectal digesta as well as a lower *Enterobacteriaceae* count in the colorectal digesta. The research showed that carcasses of the pigs who received the EM^®^Bokashi probiotic had a higher lean percentage and lower fat content than the carcasses of the control fatteners. Diet supplementation with the EM^®^Bokashi probiotic resulted in a lower pH and technological yield (TY) and a higher drip loss and shear force at a lower protein content in LL muscle. Moreover, the administration of the probiotic to fatteners resulted in higher yellowness (b*) and saturation (C*) and higher concentrations of Na, Mg and Se in meat.

## 1. Introduction

Antibiotics and chemotherapeutics have been routinely used in prophylactic dosages in animal production for decades. However, due to growing concerns related to the risks of developing cross-resistance and multiple antibiotic resistance in pathogenic bacteria in both humans and livestock, supplementation with probiotics for domestic animals has spurred huge interest in recent years [1,2]. 

Probiotics are defined as live microorganisms that improve the homeostasis of the intestinal microbiota of growing animals. Supplementation with probiotics is more effective in stimulating the immune system in piglets than in fatteners [3,4]. This effect can be linked to the maturation of the gastrointestinal tract and the immune system. Probiotics mainly act as immunostimulators through the pathways of cellular and humoral immunity, the production of immunoglobulins and interferon, the activation of macrophages, lymphocytes and NK (Natural Killer) cells and the regulation of phagocytosis [5]. These processes support maintaining the balance of Th1 and Th2 helper lymphocytes and the production of specific types of cytokines [6,7]. They are also believed to make post-weaning pigs less susceptible to intestinal disorders and to enhance their ability to digest feed components [8,9]. In addition to affecting the host’s immune system, probiotics affect the gut microbiome by altering the composition of the gut microbiota. Their most common beneficial effect in pig rearing is reduced mortality of piglets before weaning and early post-weaning [10,11]. Moreover, probiotics and prebiotics were reported to elicit a positive effect on animals’ growth performance [12]. The effects of probiotics on pig carcass and meat quality have been published in several papers, but the results have been inconclusive; most papers suggested positive effects of dietary probiotics on carcass [13] and pork quality [14,15]. The main reasons for differences in the results were found to be the specificity of the host and its interactions with the probiotic strain. This finding underlines the need for developing multi-strain preparations that would be more effective in regulating gut microbiota and/or increasing livestock growth performance. 

Previous studies postulated that the use of EM^®^Bokashi as a feed additive resulted in increased concentrations of pro-inflammatory cytokines TNF-α and IL-6, which, in turn, increase the protective capacity of colostrum by stimulating cellular immune mechanisms protecting the sow and newly-born piglets against infection. Moreover, the increased concentrations of cytokines IL-4, IL-10 and TGF-β, and of immunoglobulins in colostrum and milk, proved the immunoregulatory effect of EM^®^Bokashi on Th2 cells, leading to increased expression of regulatory T cells and the polarization of the immune response from Th1 to Th2 [16]. The results of yet other study indicated that EM^®^Bokashi supplementation had a positive effect on the morphological characteristics of porcine jejunum and caused an increase in the gene expression related to the key metabolic pathways of the gastrointestinal tract [17].

Due to the promising results cited above, pork producers have been encouraged to use EM^®^Bokashi probiotic preparations throughout the production cycle, from nursery piglets to fattening pigs. Thus far, the proposed dosage of the preparation has been aimed at achieving a desirable balance between improved pig production efficiency and its economic justification. In addition, it is also of the interest of pork production chains to explore whether the supplementation of pig diets with such a probiotic preparation will also have a positive effect on carcass and meat quality parameters.

The objective of this study was to determine the effect of EM^®^Bokashi probiotic supplementation on gut microbiota, slaughter value, technological quality and mineral content in the *longissimus lumborum* muscle of pigs.

## 2. Materials and Methods

### 2.1. Animals and Feed

In our research, we analyzed pork quality as a result of the use of the Bokashi probiotic, the dosage of which in the feed was developed and improved by the breeder, providing him the best results in terms of production efficiency and economy. The research was conducted with 120 commercial hybrid pigs (P-76 boars and Naїma sows) raised on a production farm in the Pomeranian Voivodship (Poland). During the growing period, the pigs (28 to 164 days of age) were exposed to the same environmental conditions and were fed the same balanced, dry, loose, complete feed diets ad libitum. Detailed information about diet components and feedstuff chemical composition has been previously published [18]. During the fattening period (from 78 days of age), the animals were allocated to 10 pens (each about the size of 10 m^2^), with 12 animals in each replication (aiming to keep the sex ratio at 1:1). The pen area for a single head was about 0.8 m^2^.

Pigs from the experimental group (*n* = 60) received the EM^®^Bokashi probiotic in their feed (wheat bran, sugar cane molasses (0.0785 mL/100 g), a complex of probiotics, e.g., *Saccharomyces cerevisiae* IFO 0203 3.3 × 10^5^ cfu/g; *Lactobacillus casei* ATCC^®^7469™ 1.95 × 10^7^ cfu/g; *Lactobacillus plantarum* ATCC^®^8014™ 1.95 × 10^7^ cfu/g), which was manufactured by Greendland Technologia EM® (Janowiec n/Wisłą, Poland), the authorized representative of the EM Research Organization (EMRO), Japan Technology, on the Polish territory. The probiotic supplementation varied depending on the age of the pigs, i.e., from the 28th day of life until reaching body weight of 12 kg–10 g/kg; until body weight of 20 kg–7 g/kg; until body weight of 30 kg–5 g/kg; during the fattening period, from 30 kg until the end of rearing—3 g/kg. Such a dosage of the probiotic has enabled pig producers to improve the biological efficiency of production while maintaining production profitability. The pigs in the control group (*n* = 60) did not receive the probiotic EM^®^Bokashi in their feed. The experimental and the control group pigs were not administered any antibiotics for therapeutic purposes.

After reaching the weight of approximately 112 kg, the pigs were brought to the meat processing plant on the same day and rested at lairage for approximately 16 h. They were slaughtered the next morning after a cumulative period of ante-mortem fasting of 33 h 30 min.

### 2.2. Carcass and Meat Quality

All the experimental animals (*n* = 60; an equal number of gilts and barrows) were slaughtered at a commercial abattoir that operated a Butina CO_2_ gas stunning system (Marel, Garðabær, Iceland). The longissimus dorsi’s initial pH was measured at 35 min after stunning with a portable CP-411 pH-meter (Elmetron, Zabrze, Poland). Carcass lean percentage was estimated by a CGM optic-needle apparatus (Sydel, Lorient, France) using standard measurements of backfat thickness and the depth of m. longissimus dorsi muscle between the 3rd and 4th ribs, 7 cm from the mid-line, on the left half-carcass. Hot carcass weight was also determined at 35 min post mortem. 

The carcasses were chilled using a two-stage blast and conventional chilling system. Directly from the evisceration line, the carcasses were chilled in a blast-cooling tunnel at the operating temperature of −24 °C for 70 min. Subsequently, the carcasses were placed in an equilibration cooler operating at 1 °C for 22 h. In the equilibration cooling room, 30 carcasses of a similar weight (HCW: 85 ± 5 kg) and sex (1:1) were selected from the probiotic-supplemented group and 30 were selected from the control group to determine meat quality traits.

At 24 h post mortem (p.m.), pH was measured using the aforementioned pH-meter, and electrical conductivity (EC_24_) was established by using an LF-Star (Ingenieurbüro Matthäus, Nobitz, Germany). The 24 h post-mortem muscle pH and EC were measured between the 4th and 5th lumbar vertebrae of the right half carcasses. 

The *longissimus lumborum* (LL) muscle samples containing backfat layers were collected between the 1st and 4th lumbar vertebral regions of the right half-carcasses. The samples were wrapped in aluminum foil and transported for 1 h to the laboratory in vacuum flasks. Immediately upon arrival to the laboratory, the LL muscle was separated from the fat and bones. 

Subsequently, all LL muscle samples were cut into 4 cm thick slices (starting from the cranial end) and placed in polyethylene bags for drip loss, technological yield (TY), pH and color determinations. Drip loss was established as a percentage of weight loss after 1 day (48 h p.m.) of storage at 4 °C, according to Prange et al. [19]. At 48 h p.m., meat and backfat color, i.e., lightness (L*), redness (a*), yellowness (b*) and chroma (C*), were measured on freshly cut surface muscle and backfat after 20 min of blooming at 4 °C with a HunterLab Mini Scan XE Plus 45/0 (HunterLab Inc., Reston, VA, USA) containing a standard illuminant D65 and 10° Standard Observer. 

Meat yield during curing and thermal processing (72 °C), expressed by a TY (technological yield) indicator, was determined according to Naveau et al. [20], as modified by Koćwin-Podsiadła et al. [21]. The samples of LL muscle were taken 24 h after slaughter. Meat cubes (1 × 1 × 1 cm) were immersed in a solution containing 12% NaCl, 0.07% NaNO_2_ and 0.06% glucose. After 24 h of curing at 4 °C, the samples were thermally processed in a water bath (to an internal temperature of 72 °C). The remaining part of the left loin was vacuum-packed in polyethylene bags and frozen at −19 °C for a maximum of 2 months until proximate analysis and mineral content determination.

### 2.3. Shear Force 

The thawed (at 4 °C for about 24 h) LL muscle samples (about 200 g) were heated in a plastic bag in water at 80 to 81 °C until reaching an internal temperature of 72 °C, and next cooled to 20 °C. Shear force was measured using a Warner–Bratzler apparatus (WB) manufactured at the Baking Industry Research Centre (Bydgoszcz, Poland). Cylindrical meat samples, cut out (along muscle fibers) with a cork borer 1.0 cm in diameter, were placed in a triangular recess under five blades of the tenderness measuring instrument, which recorded the maximum force (expressed in kilograms) required for cutting through the meat. The final result presented for each sample was the average of three consecutive measurements.

### 2.4. Proximate Analysis

The following chemical analyses were performed on the experimental and control ground muscle samples (AOAC methods [22]: (1) moisture content was estimated on 2 g samples at 102 °C; (2) crude protein content was estimated using a standard macro-Kjeldahl method and (3) intramuscular fat content was established by petroleum ether extraction using a Soxhlet apparatus). 

### 2.5. Mineral Composition

The concentrations of selected meat and feed micro- and macro-elements were determined by emission spectrometry with excitation in inductively coupled argon plasma (ICP OES) using an Optima 2000 DV apparatus (PerkinElmer Inc., Boston, MA, USA). Samples for the spectrometric analysis were mineralized in a microwave system (Anton Paar, Graz, Austria). From the homogenized meat samples, aliquots of 0.6 g were made, placed in pressure quartz vessels and then 5.0 mL 65% HNO_3_ and 0.5 mL 30% H_2_O_2_ (Suprapur®, Merck KGaA, Darmstadt, Germany) were added. Closed vessels were placed in a mineralizer equipped with a continuous temperature and pressure control system. The solutions were left for about 20 min for CO_2_ and NO_2_ volatilization; this was followed by making up to 10 mL in volumetric flasks. The concentrations of selected microelements were directly determined in the solutions prepared, Cr, Mn, Fe, Cu, Zn, while for the determination of the macroelements Na, K, Ca, Mg and P, the solutions were 10-fold or 100-fold diluted to obtain optimal ranges of spectrometer concentration. Emission measurements for microelements were carried out when choosing a longer axial optical path, while macronutrients were analyzed radially, across the plasma. The standard for analysis was the certified ICP Multielement Standard IV from Merck.

Selenium concentrations in LL muscle samples were determined with a spectrofluorometric method using a Shimadzu RF-5001 PC analyzer. The samples (1 mL) were digested in HNO_3_ at 230 °C for 180 min and then in HClO_4_ at 310 °C for 20 min. Afterwards, 9% HCl was added to the digested samples to reduce selenate VI to selenate IV. Subsequently, selenate IV was complexed with 2,3-diaminonaftalene (Sigma-Aldrich®, Merck KGaA, Darmstadt, Germany), and the resulting complex was extracted with cyclohexane. The excitation wavelength was 376 nm, and the fluorescence emission wavelength was 518 nm. 

### 2.6. Microbiological Determinations

The constant supplementation of the probiotic of 3 g/kg of feed enabled the microbiological sampling to be uniformly spaced throughout the duration of the experiment. Duplicate samples of the two experimental feeding regiments (T1: from 45 to 65 kg body weight; T2: 65 kg until the end of the fattening period) were collected immediately after the mixing of ingredients from the probiotic-supplemented and the control group for microbiological analyses. During the test, two samples were taken every two weeks from each feed and the obtained results were averaged. The samples of proximal colon sections were collected for microbiological tests as described in our previous study [18].

Official standard methods were used to determine specific groups of microorganisms: total bacterial count (TBC)—[23]; total yeast and mold count (TYMC)—[24]; total count of *Enterobacteriaceae* (TCE), capable of degrading trichloroethylene—[25]; lactic acid bacteria (LAB)—[26]; anaerobic spore-forming bacteria—Clostridium (CL)—[27] and pathogenic bacteria, *Salmonella* spp. [28] and Listeria monocytogenes—[29]. The protocol described in [30] was followed in order to ensure the reliability of microbiological tests in relation to particular standards. Isolated bacterial and yeast strains were subjected to a diagnostic analysis considering biochemical features using API 50 CHL and API ID 32 C tests (BioMerieux Inc., Marcy-l’Étoile, France).

#### Identification of LAB and Yeasts

Species affiliation of all isolated strains, initially classified to LAB and yeast, was confirmed with the PCR technique. Cultures of *Lactobacillus* were incubated on the de Man Rogosa Sharpe Broth (1.0% proteose peptone, 0.8% malt extract, 0.4% yeast extract, 2.0% glucose, 0.5% sodium acetate, 0.2% triammonium citrate, 0.02% magnesium sulfate, 0.005% manganese sulfate, 0.2% dipotassium phosphate, 0.1% polysorbate 80; (Scharlab S.L., Barcelona, Spain) at 37 °C for 24 h. In the case of yeast, cultures were incubated on the Yeast Peptone Broth (1.0% yeast extract, 2.0% peptone, 2.0% glucose (Scharlab S.L.) at 22 °C for 24 h. Genomic DNA was isolated following the protocol of the Genomic Mini AX Bacteria (A&A Biotechnology, Gdańsk, Poland) using mutanolisine (Sigma-Aldrich®, Merck KGaA, Darmstadt, Germany) and Genomic Mini AX Yeast (A&A Biotechnology) using lyticase (Sigma-Aldrich®). Extracted DNA was amplified using the following primers: 27F (5′-AGAGTTTGATCCTGGCTCAG-3′) and 1492R (5′-GGTTACCTTGTTACGACTT-3′) for bacteria [31], and ITS 1 (5’-TCC GTA GGT GAA CCT GCG G-3’) and ITS 4 (5’-TCC TCC GCT TAT TGA TAT GC-3’) for yeasts [32]. The PCR reaction was conducted in 25 µL of the reaction mixture containing: 10.0 µL of MIX PCR (A&A Biotechnology), 1.0 µL of each primer and 2.0 µL of DNA template. The PCR reaction was described in an earlier study by Rybarczyk et al. [33].

### 2.7. Statistical Analysis

Statistical analysis was performed to compare carcass and meat quality traits and the microbiological data between the different groups using the two-way (the fixed effect of group and gender) analysis of variance (Statistica 13.1 PL statistical package). The results of microbiological analyses were considered as the total number of microorganisms, expressed in log colony forming units. A detailed comparison of the means was analyzed with the Tukey’s test at *p* ≤ 0.01 and *p* ≤ 0.05. The tables present average values and their standard errors. The levels of similarity between the microbial profiles of individual samples were determined on the basis of the cluster analysis using Ward’s method. 

## 3. Results

### 3.1. Microbiological Analysis

The results of the microbiological analyses of feed samples and colon sections collected from the pigs did not indicate *Salmonella* spp. and *L. monocytogenes*. The analysis of microbiological results showed significant differences in the population numbers between the determined microorganisms: LAB, TYMC and TBC in all tested samples (Table 1). Microorganisms belonging to the genera *Lactobacillus* and *Saccharomyces* were found in all analyzed samples of the feed and the probiotic preparation EM^®^Bokashi, whereas yeasts belonging to the genus *Candida* were determined sporadically. In the case of EM^®^Bokashi probiotic and the feed + EM^®^Bokashi mixture, the microbial profile was predominated by *S. cerevisiae* and *L. casei*. In the case of *L. plantarum*, the number of these microorganisms was determined at a lower level (EM^®^Bokashi probiotic and feed + EM^®^Bokashi probiotic).

The evaluation of differences in the population numbers of microorganisms determined in the analyses showed that LAB and TYMC were statistically more often isolated from the mucosa of individuals supplemented with EM^®^Bokashi probiotic than in the samples from the fatteners from the control group. An opposite trend of changes was determined for anaerobic bacteria (*Clostridium*), the count of which was statistically lower in the samples taken from individuals supplemented with the probiotics (Table 2). In the case of microorganisms isolated from the chyme, there were no statistically significant differences in the number of isolated LAB between the groups of pigs. However, the study showed that EM^®^Bokashi probiotic supplementation resulted in a significantly higher count of yeasts and a lower count of *Clostridium* and *Enterobacteriaceae* in the colorectal digesta.

The overall microbiological profile of the studied groups was also analyzed. Based on cluster analysis, it was found that the structure of the number of microorganisms isolated from pigs administered the feed that had been supplemented or non-supplemented with EM^®^Bokashi probiotic was statistically different (Figure 1). The level of similarity between the quantitative structure of microorganisms colonizing the intestinal mucosa and present in the chyme in individual groups of fattening pigs did not differ significantly. In the case of animals supplemented with the EM^®^Bokashi probiotic, the statistical analysis showed a similarity at the level of 72%, while in the control group this was 42%; the correlation coefficients were 0.850 and 0.468, respectively.

### 3.2. Carcass and Meat Quality

The research showed that the carcasses of pigs who received the EM^®^Bokashi probiotic had a higher lean percentage, thicker *longissimus dorsi* muscle and thinner backfat, with a hot carcass weight similar to the control animals (Table 3). The proximate chemical composition of the LL muscle showed no significant differences between the analyzed groups (Table 4), except for the total protein, which was significantly higher in the meat of fattening pigs from the control group than those supplemented with the EM^®^Bokashi probiotic.

The LL muscle of the fatteners supplemented with the EM^®^Bokashi probiotic had a significantly lower pH at 24 h p.m. (pH_24_) and a lower technological yield (TY) compared to the control group (5.74 and 94.67%, respectively). Moreover, the fattening pigs that were administered probiotics in the feed had a significantly increased drip loss and shear force compared to the control group.

Statistical analysis revealed a significant difference between the analyzed groups regarding color characteristics (b*, C*) of the LL muscle. The pigs supplemented with the EM^®^Bokashi probiotic had higher values of yellowness (b*) and chroma (C*) color parameters compared to the control animals. Furthermore, no significant differences were found in the characteristics of the backfat color (Table 5) between the fatteners of the analyzed groups, with the exception of hue angle (*h^o^*), which was significantly higher for the backfat obtained from control fattening pigs compared to the animals supplemented with the probiotic.

### 3.3. Mineral Concentration

The research showed that the LL muscles of pigs who received the EM^®^Bokashi probiotic had a significantly higher concentration of Na, Mg and Se compared to the muscles of the control fatteners (Table 6).

## 4. Discussion

Despite the lack of statistically significant differences between the total number of microorganisms isolated from the analyzed groups of fatteners, the fact of the increased adherence of LAB from EM^®^Bokashi-supplemented feed to proximal colon mucosa becomes important. The natural features of selected and probiotic LAB predispose these microorganisms for competitive displacement of other species [34]. As a result of the colonization of the environment, the health-promoting mechanisms of the higher organisms are stimulated and the accompanying microbiota’s diversification changes [18,33]. The effects of microbiological conversion are also observed at the level of technological properties of the obtained raw material. It should be noted here that they are not always desirable [18].

Based on the microbiological analysis of the material collected during the study, it was shown that the addition of the EM^®^Bokashi probiotic preparation to the feed firstly increased the number of LAB and reduced the number of *Clostridium* in the colon mucosa. In the case of the colon, such a relationship was also observed for the bacteria of the *Enterobacteriaceae* family, the number of which was lower. These dependencies are primarily due to LAB’s capability for displacing competing bacteria. The mechanisms of these processes’ regulation are, however, complex. One of the attributes of lactic acid fermentation bacteria is the ability to displace such microorganisms as *Salmonella*, *E. coli* and *Clostridium* spp. by reducing their adhesion to the intestinal mucosa [Dowarah et al. [12] and Giang et al. [35]. Moreover, Zhang et al. [36] described a similar dependency, showing that supplementing diets with *Lactobacillus rhamnosus* significantly reduced the number of *E. coli* bacteria in the colon. According to the authors, it had a direct impact on the improvement in the health of piglets and they observed a reduced occurrence of diarrhea. Huang et al. [37] and Chiang et al. [38] claim that population number regulation by *Lactobacillus* is also significantly affected by hydrogen peroxide and lactoferrin, which may show antagonistic activity against *E.*
*coli* and *Enterobacteriaceae*. The modification of piglet gut microbiome upon LAB strains results in the enhanced proliferation of *Lactobacillus* and *Bifidobacterium*. Moreover, Lactobacilli added as a probiotic to the diet stimulated beneficial fermentation, which subsequently increased the concentrations of short chain fatty acids and lactic acid in the GIT. Investigations conducted by Kim et al. [39] demonstrated that the metabolic activity of microorganisms classified into this group also contributes to an increase in the absorption of nutrients in the intestine. The histometric analysis of the jejunum conducted by Dowarah et al. [12] showed that supplementation with *P. acidilactici* FT28 and *L. acidophilus* NCDC-15 strains in feed had an effect on increasing the villus height and the crypt depth in the pigs supplemented with probiotics compared to the control group. Such observations were also made by Suo et al. [14] after the application of the *L. plantarum* ZJ316 probiotic strain in the feeding of fatteners. They also found that there may be other mechanisms by which probiotics alter the permeability of the intestinal epithelium and elongate the intestinal villi. The effect of the *L. plantarum* ZJ316 probiotic on pig growth was closely related to the dose of the preparation, as evidenced by the fact that the use of a lower dose (1 × 10^9^ cfu/g) was significantly associated with higher weight gain of the pigs compared to the higher probiotic dose (5 × 10^9^ cfu/g and 1 × 10^10^ cfu/g), which may be related to the functioning of the host’s immune system. Suo et al. [14] explain this phenomenon by the fact that the higher the dose of the probiotic we use, the more we influence the immune system response and, at the same time, that this higher dose may deteriorate the production results of the fatteners.

In our study, the pigs supplemented with the EM^®^Bokashi probiotic had carcasses with a better musculature and, at the same time, less fat. In a study by Barowicz et al. [40], a 0.35% addition of the Acid Pack 4 Way probiotic (*B. subtilis*, *L. acidophilus*, *E. faecium*) did not influence pig carcass traits. Furthermore, Jukna et al. [41] found no effect of Yeasture (*S. cerevisiae*, *L. casei*, *L. acidophilus*, *S. faecium*, *B. subtilis*) and Microbond (*S. cerevisiae*, *L. acidophilus*, *S. faecium*, *B. subtilis*, selenium, chromium) on the carcass weight, slaughter yield or the chemical composition of meat, except for the carcass output, which was 2.0 to 2.1% higher in the groups supplemented with those probiotics when compared to the control groups. According to He et al. [42], the intestinal microbiome has a significant influence on the fatness of the pig carcass due to the ability of the bacteria of the genera *Lachnospiraceae*, *Ruminococcaceae*, *Prevotella*, *Treponema* and *Bacteroides* to produce short-chain fatty acids by fermenting non-digestible polysaccharides and pectins. Short-chain fatty acids can regulate energy homeostasis in the host, protect the host from inflammation and inhibit the development of adipose tissue. If the balance of the microbiome fraction, including the above-mentioned bacteria, is disturbed, this may result in an increased fatness of pigs. In broiler chickens, Wen et al. [43] showed a significant contribution of the caecum and duodenum microbiota in fat deposition, which is related to the activity of microorganisms in the field of hydrogen and formate reduction and polysaccharide fermentation, significantly increasing the amount of products such as acetate, propionate and butyrate in the intestines and allowing the organism to absorb more nutrients and energy. Recent studies have shown that the application of the probiotic *L. reuteri* strain significantly reduced the diameter of the muscle fibers and the cross-sectional area of the *longissimus thoracis* muscle [44].

In this study, a poorer quality of meat was found as a result of supplementation with the EM^®^Bokashi probiotic. This is inconsistent with the results of some authors who observed no effect of probiotics on physicochemical traits [41]. On the other hand, in the study by Jukna et al. [41], the use of Yeasture and Microbond probiotics did improve the culinary properties of LL muscle; cooking loss decreased by 5.4 to reach 6.1%, water holding capacity increased by 1.8 to 3.2%, and meat hardness decreased by 6.9 to 47.2%. In addition, Liu et al. [15] showed that supplementation with probiotics (yeasts, lactic acid-producing bacteria and *Bacillus subtilis*) reduced the drip loss and cooking loss of pork but had no effect on the pH or shear force. However, Chang et al. [45] found that the free leakage from the muscles of fatteners supplemented with probiotics (*L. plantarum*) was significantly greater at a lower pH, but without an impact on WHC (water holding capacity). Moreover, Rybarczyk et al. [18] found a poorer meat quality in pigs supplemented with the liquid probiotic EM^®^, and this unfavorable effect was caused by the higher dose of the probiotic used. However, the research by Khanal et al. [46] proves that the composition of the gastrointestinal microbiota was not related to the majority of the analyzed meat quality traits, which suggests that the composition of the intestinal microbiota should mainly be treated as an environmental factor that may be subject to modifications. 

In our research, based on the analysis of LL muscle color characteristics, it was found that the pigs supplemented with the EM^®^Bokashi probiotic had higher values of color chromatic characteristics, i.e., yellowness (b*) and chroma (C*), compared to the control group. Additionally, our other study [18] showed that higher b* and C*, but also higher redness (a*), were found at the highest dose of the EM probiotic in the feed (0.5%) compared to the control group of pigs. In turn, Li and Chen [47] found that the addition of probiotics to feed (*L. acidophilus*, *S. cerevisiae*, *B. subtilis*) was significantly related to the higher color stability of the meat of the pigs. In the research by Jiang [48], the addition of a probiotic preparation containing *Phaffia rhodozyma* significantly increased the redness (a*) of fatteners’ meat color. Other investigations have shown that the muscles of pigs with a higher proportion of type I fibers and a lower proportion of type II fibers were characterized by a higher redness (a*) of meat color, which was related to a higher concentration of myoglobin [49,50]. Tian et al. [46] showed the effect of *L. reuteri* probiotic supplementation on the reduction in muscle fiber diameter and the cross-sectional area of the *longissimus thoracis* muscle, which had a beneficial effect on the color and, in particular, on the redness (a*). In other studies, it was shown that the administration of probiotics (*L. plantarum*) to pigs enhanced the antioxidant activity in meat, which was due to an increase in the concentration of vitamin C [45]. It should be mentioned here that vitamin C is characterized by very good antioxidant properties and increasing its concentration in meat improves meat color and persistence [51].

In the present study, it was shown that supplementing the feed of fatteners with the EM^®^Bokashi probiotic resulted in a higher concentration of sodium, magnesium and selenium in their meat. This may be due to the ability of the LAB regarding the biotransformation of the inorganic selenium in vitro, ex vivo and in vivo. The biotransformation of inorganic selenium consumed with feed or litter elements into organic selenium (i.e., into selenoamino acids) may directly translate into its bioavailability and absorption, which may affect its concentration in meat [52]. In unpublished own research, and related to the experimental system presented in the publication by Rybarczyk et al. [18], it was found that the meat of pigs supplemented with the highest dose of liquid probiotic EM^®^ (0.5%) had the highest concentration of magnesium. In research conducted with the in vivo models, Chang et al. [45] proved that the concentration of magnesium and potassium was higher in the meat of pigs supplemented with a probiotic, but these noted differences were not statistically significant. These authors also found that the content of calcium, iron and zinc was lower in the group of fatteners supplemented with probiotics compared to the control group. Skrypnik and Suliborska [53] claim that changes in the intestinal microbiome may directly translate into the bioavailability and absorption of minerals. Current research shows that probiotic supplementation can have a significant impact on the absorption and metabolism of calcium and phosphates.

In our research, the fatteners supplemented with the EM^®^Bokashi probiotic were characterized by higher carcass fleshiness and, at the same time, poorer quality of meat. The obtained results may confirm the negative correlation found in other studies between the degree of muscularity of the carcass and the quality of the meat, which is also often due to a reduction in the content of intramuscular fat [54,55]. Moreover, our research showed a significantly lower pH_24_ and technological yield and an increased drip loss and shear force of the LL muscle in pigs receiving the EM^®^Bokashi probiotic. As indicated by Huff-Lonergan et al. [56], these characteristics are closely connected. In their research, pH_24_ significantly negatively correlated with the drip loss (−0.33) and positively correlated with tenderness (0.27). On the other hand, drip loss significantly positively correlated with shear force (0.29 and 0.34). Moreover, Miar et al. [57] showed high negative genetic correlations between drip loss and pH and shear force. 

## 5. Conclusions

Our results indicated that dietary supplementation of pig finishing diets with an EM^®^Bokashi probiotic preparation containing *L. casei, L. plantarum* and *S. cerevisae* increased the intestinal counts of LAB and yeasts. At the same time, we observed a noticeable decrease in the counts of fecal Enterobacteria and Clostridia, the pathogens that are typically responsible for diarrheal diseases in pigs. 

This change in the microbiota composition had a positive influence on carcass quality traits and on the concentration of Mg and Se in the meat of the experimental pigs but simultaneously had a negative impact on the technological quality of the meat. By influencing gut microbiota, the LAB and yeasts dosage could trigger changes in LL muscle quality, especially affecting the traits associated with water holding capacity and meat color characteristics (b*, C*). Therefore, we concluded that the use of the EM^®^Bokashi probiotic at a dose of 3 g/kg of feed during the last stage of pig production is not practical. Alternatively, its dose should be reduced.

## Figures and Tables

**Figure 1 animals-11-03590-f001:**
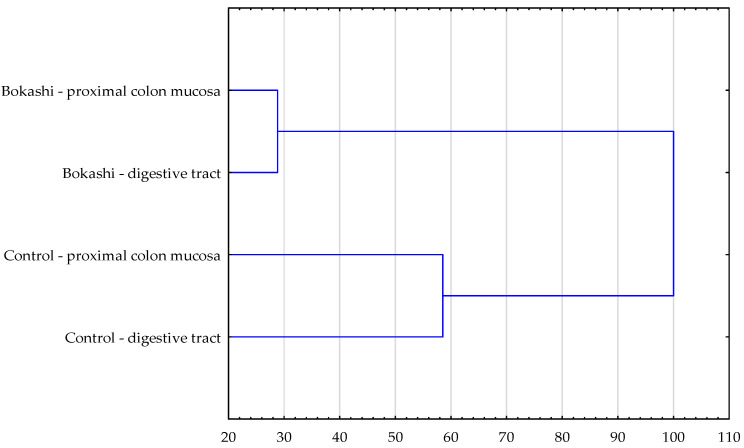
Dendrogram of the cluster analysis of the similarity of the microbiological profile in the studied groups.

**Table 1 animals-11-03590-t001:** Microbiological composition of feed.

Microbiota Composition (log10/g)		Control	Feed + Bokashi Mixture	Bokashi (Preparation)	*p*-Value
LAB	*L. plantarum*	0.95 ^c^ ± 0.18	1.92 ^b^ ± 0.06	2.81 ^a^ ± 0.35	0.001
	*L. casei*	1.54 ^c^ ± 0.07	3.73 ^b^ ± 0.29	4.68 ^a^ ± 0.35	0.005
	Total	1.90 ^c^ ± 0.18	3.92 ^b^ ± 0.06	4.96 ^a^ ± 0.35	0.003
TYMC	*S. cerevisiae*	0.25 ^c^ ± 0.33	3.98 ^b^ ± 0.25	6.88 ^a^ ± 0.08	0.009
	*Candida* spp.	< 1.0	<1.0	1.0 ± 0.05	-
	Total	0.25 ^c^ ± 0.33	3.98 ^b^ ± 0.25	6.88 ^a^ ± 3.14	0.003
TBC	-	3.28 ^a^ ± 0.09	2.00 ^ab^ ± 0.41	1.41 ^b^ ± 0.29	0.002

Mean values in rows marked with different letters differ significantly: ^a–c^: *p* ≤ 0.05. LAB—lactic acid bacteria; TYMC—total yeast and mold count; TBC—total bacterial count.

**Table 2 animals-11-03590-t002:** Composition of colonic microbiota.

Microbiological Fractions (log10/g)	Control	Bokashi	*p*-Value
	Proximal colon mucosa		
TBC	7.90 ± 0.43	7.79 ± 0.40	0.706
LAB	6.04 ^b^ ± 0.45	7.21 ^a^ ± 1.06	0.036
TYMC	2.15 ^b^ ± 0.06	2.85 ^a^ ± 0.20	0.026
TCE	7.24 ± 1.21	6.35 ± 0.33	0.195
CL	2.77 ^a^ ± 0.01	2.27 ^b^ ± 0.28	0.009
	Digestive tract of proximal colon		
TBC	7.08 ± 0.02	7.48 ± 0.90	0.414
LAB	8.28 ± 0.44	8.18 ± 0.24	0.633
TYMC	1.79 ^b^ ± 0.01	2.24 ^a^ ± 0.28	0.016
TCE	6.36 ^a^ ± 0.05	4.61 ^b^ ± 0.56	0.001
CL	2.44 ^a^ ± 0.47	1.58 ^b^ ± 0.01	0.028

Mean values in rows marked with different letters differ significantly: ^a, b^: *p* ≤ 0.05. TBC—total bacterial count; LAB—lactic acid bacteria; TYMC—total yeast and mold count; TCE—total count of *Enterobacteriaceae*; CL—total count of *Clostridium*.

**Table 3 animals-11-03590-t003:** Slaughter carcass value.

Traits	Control	Bokashi	*p*-Value
HCW (kg)	85.30 ± 0.95	85.07 ± 0.70	0.844
Meatiness (%)	54.75 ^B^ ± 0.49	57.01 ^A^ ± 0.46	0.000
Backfat thickness (mm)	19.64 ^A^ ± 0.63	16.54 ^B^ ± 0.67	0.000
Muscle thickness (mm)	57.88 ^b^ ± 1.49	60.34 ^a^ ± 0.96	0.022

Mean values in rows marked with different letters differ significantly: ^A, B^: *p* ≤ 0.01; ^a, b^: *p* ≤ 0.05. HCW—hot carcass weight. The number of the insertions of fatteners in each group—60.

**Table 4 animals-11-03590-t004:** Meat quality and proximate chemical composition.

Traits	Control	Bokashi	*p*-Value
pH_35min_	6.63 ± 0.03	6.63 ± 0.03	0.883
pH_24_	5.74 ^A^ ± 0.05	5.58 ^B^ ± 0.02	0.003
Drip loss (%) EC_24_ (mS/cm)	2.50 ^B^ ± 0.18	4.14 ^A^ ± 0.26	0.000
	4.32 ± 0.17	3.95 ± 0.20	0.153
TY (%)	94,67 ^A^ ± 1.10	87,52 ^B^ ± 0.60	0.000
L*	55.45 ± 0.57	56.04 ± 0.53	0.449
a*	5.48 ± 0.14	5.73 ± 0.14	0.209
b*	13.89 ^b^ ± 0.14	14.37 ^a^ ± 0.14	0.014
C*	14.94 ^b^ ± 0.16	15.49 ^a^ ± 0.14	0.011
*h^0^*	68.53 ± 0.45	68.27 ± 0.52	0.705
Shear force (kg)	4.55 ^B^ ± 0.15	5.35 ^A^ ± 0.14	0.000
Total protein (%)	24.30 ^A^ ± 0.06	24.03 ^B^ ± 0.08	0.009
Intramuscular fat (%)	1.42 ± 0.09	1.46 ± 0.08	0.722
Dry matter (%)	26.08 ± 0.10	25.82 ± 0.10	0.069

L*—lightness; a*—redness; b*—yellowness; C*—chroma; *h^0^*—hue angle; EC—electrical conductivity; TY—technological yield. Mean values in rows marked with different letters differ significantly: ^A, B^: *p* ≤ 0.01; ^a, b^: *p* ≤ 0.05. The number of the insertions of fatteners in each group—30.

**Table 5 animals-11-03590-t005:** Color traits of backfat.

Traits	Control	Bokashi	*p*-Value
L*	83.98 ± 0.44	83.23 ± 0.23	0.138
a*	2.76 ± 0.14	3.09 ± 0.15	0.098
b*	11.41 ± 0.22	11.00 ± 0.21	0.182
C*	11.76 ± 0.23	11.45 ± 0.21	0.334
*h^0^*	76.43 ^a^ ± 0.61	73.98 ^b^ ± 0.75	0.014

L*—lightness; a*—redness; b*—yellowness; C*—chroma; *h^0^*—hue angle. Mean values in rows marked with different letters differ significantly: ^a, b^: *p* ≤ 0.05. The number of the insertions of fatteners in each group—30.

**Table 6 animals-11-03590-t006:** Contents of macro- and micro-elements in feed and meat.

Traits (mg/kg)	Feed	Control	Bokashi	*p*-Value
K	5216	2941.55 ± 16.17	2941.77 ± 18.97	0.993
Na	1414	346.30 ^b^ ± 4.18	361.81 ^a^ ± 4.73	0.017
Mg	1615	266.40 ^b^ ± 1.09	271.22 ^a^ ± 1.57	0.014
P	4909	2203.53 ± 10.76	2217.70 ± 11.27	0.367
Cr	0.12	0.04 ± 0.00	0.05 ± 0.00	0.087
Mn	81.7	0.09 ± 0.00	0.08 ± 0.00	0.538
Fe	203	5.21 ± 0.29	5.07 ± 0.11	0.662
Cu	20.1	0.40 ± 0.01	0.41 ± 0.01	0.456
Zn	106.2	13.74 ± 0.19	13.45 ± 0.20	0.318
Ca	6537	36.18 ± 0.46	37.44 ± 0.52	0.076
Se	0.18	0.08 ^B^ ± 0.00	0.09 ^A^ ± 0.00	0.001

Mean values in rows marked with different letters differ significantly: ^A, B^: *p* ≤ 0.01; ^a, b^: *p* ≤ 0.05. The number of the insertions of fatteners in each group—30.

## Data Availability

The data presented in this study are available on request from the corresponding author.

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
