# Peer review of "Carcass and Pork Quality and Gut Environment of Pigs Fed a Diet Supplemented with the Bokashi Probiotic"

_animals, 2021, doi:10.3390/ani11123590_

Round 1

Reviewer 1 Report

I have no further comments on this manuscript, the Authors have complied with the comments. The work is suitable for printing.

Author Response

Thank you.

Reviewer 2 Report

General comments

  1. In the materials and methods, it appears the probiotic was fed to the pigs from d 28 until slaugther. How is this economically viable? considering that the value of better carcass loin with not be cost effective. Please explain why the feeding of the probiotic was done that way.
  2. Why didnt you focus on feeding the probiotics in early life of the piglets and then follow them to market to determine the impact. since you already concede that the effects of probiotics are more pronounced in the piglets? see your L42-44
  3. You explain in L180-184 how the microbiological sampling was done, but you failed to mention at what point in the trial this was done. THis is important because each phase of the trial had a different amount of probiotic per tonne of feed.
  4. Considering that the study was done in phases of feeding the probiotics, the statistical analysis section should reflect how this was done, unfortunately that is not the case in the description given in the stats,. please update that section with more details.
  5.  Generally, there was no linkage explaining the mode of action of the probiotics in relation to the impacts on carcass parameters. In other words the paper did not carry a message explaining how probiotics in finisher pigs impacts carcass.
  6. It was not clear in your explanation how the selected probiotics could influence mineral content in the meat in your discussion L407-424.
  7. It will be inaccurate the make the conclusions you make no clear inference when the gut microbiota data was taken. This will have influence on the effects on growth performance.

overall comments

Authors are advised to present more details in the materials and methods section and statistical analysis section to be clear on the desgin of the study in order to aid in interpretation.

Author Response

  1. In the materials and methods, it appears the probiotic was fed to the pigs from d 28 until slaugther. How is this economically viable? considering that the value of better carcass loin with not be cost effective. Please explain why the feeding of the probiotic was done that way.

The effective amount of supplementation with the Bokashi probiotic, depending on the growth stage of the pigs, has been developed by the breeder. The developed technique of using a probiotic by the breeder allows him to achieve the best results of fattening efficiency and production profitability.

  1. Why didnt you focus on feeding the probiotics in early life of the piglets and then follow them to market to determine the impact. since you already concede that the effects of probiotics are more pronounced in the piglets? see your L42-44.

In our research, the goal was to determine whether a probiotic affects the final consumer product, i.e. the quality of the carcass and meat. In future research on the Bokashi probiotic, we will also consider production and health outcomes throughout the production cycle.

  1. You explain in L180-184 how the microbiological sampling was done, but you failed to mention at what point in the trial this was done. THis is important because each phase of the trial had a different amount of probiotic per tonne of feed.

In our research, we actually analyzed the fattening period when the amount of probiotic administered was kept at a constant level (from 30 kg until the end of rearing - 3 g / kg). The material collected for microbiological tests was also related to this probiotic dosage.

  1. Considering that the study was done in phases of feeding the probiotics, the statistical analysis section should reflect how this was done, unfortunately that is not the case in the description given in the stats,. please update that section with more details.

In our study, we did not analyze the production results at individual stages of pig growth with the change in the dosage of the Bokashi probiotic. The final quality of the carcass and meat was analyzed depending on the method of supplementing with the Bokashi probiotic.

  1.  Generally, there was no linkage explaining the mode of action of the probiotics in relation to the impacts on carcass parameters. In other words the paper did not carry a message explaining how probiotics in finisher pigs impacts carcass.

In part L. 354-375 we indicated what could have caused the fattening pigs supplemented with probiotics to have higher meatiness and lower fat thickness.

  1. It was not clear in your explanation how the selected probiotics could influence mineral content in the meat in your discussion L407-424.

In part L. 413-430 we indicated what could have caused that fattening pigs supplemented with probiotics had a higher concentration of certain micro- and macroelements.

  1. It will be inaccurate the make the conclusions you make no clear inference when the gut microbiota data was taken. This will have influence on the effects on growth performance.

We took feed samples for microbiological tests with the same dosing (3 kg/tone) of the feed probiotic. Appropriate information in this regard has been introduced in the methodology of microbiological tests. Let me remind you that our research concerned the final quality of the carcass and meat, not the fattening efficiency, so the increased dosing of the probiotic after weaning was not included in the statistical methods of the analysis of the results. It should be emphasized here that we did not analyze growth performance in the research.

Reviewer 3 Report

I found this work well done and well described. there are only minor comments that I add below:

Line 87: please insert the number of pig into experimental group

Line 103: how did you select the carcasses? Please explain

Line 135: what “atn” is?

194: you should add 2.6.1 “prima” identification LAB

Line 233: first LAB, then TYMC and last TBC

Line 252: delete “-“

Line 255: in the table note please follow the sequence of item

Line 345-347: please reword this sentence

Line 372: “phisicco”? are you sure it is correct?

Line 382: reword the sentence in “this unfavorable effect was caudes by the higher dose of the pro-382 biotic used.”

Line 384: the mayority of what? Please explain better this concept

Line 394: please change in “persistence in pigs’ meat”

Line 444: “influenced carcass”; after “negatively” you should insert a verb

Author Response

I found this work well done and well described. there are only minor comments that I add below:

Line 87: please insert the number of pig into experimental group. Added in the text.

Line 103: how did you select the carcasses? Please explain. Corrected in the text.

Line 135: what “atn” is? Corrected. Should be „at”.

194: you should add 2.6.1 “prima” identification LAB. Added 2.6.1.

Line 233: first LAB, then TYMC and last TBC. Corrected.

Line 252: delete “-“ Corrected.

Line 255: in the table note please follow the sequence of item. Corrected.

Line 345-347: please reword this sentence. The sentence corrected.

Line 372: “phisicco”? are you sure it is correct? Corrected. Should be „physico”

Line 382: reword the sentence in “this unfavorable effect was caudes by the higher dose of the probiotic used.” OK, corrected.

Line 384: the mayority of what? Please explain better this concept. Sentence corrected.

Line 394: please change in “persistence in pigs’ meat”. OK. We change on „stability” in the text.

Line 444: “influenced carcass”; after “negatively” you should insert a verb. Corrected.

Reviewer 4 Report

Comments, review of manuscript id: Animals-1459223.

Title: ”Carcass and pork quality, and gut environment of pigs fed a diet supplemented with the Bokashi probiotic”

The manuscript describes the use of a probiotic supplement in feed for pigs, and its effect on gut health and pork quality. The manuscript needs some corrections, as mentioned in the comments below.  

The English language need some corrections.

Comments.

General comments. Some places in the manuscript you write “EM Bokashi”, and some other places you use “EM®Bokashi”. Please be consequent in how you write this, I prefer “EM®Bokashi”.

Abstract, line 29. When you write “meat percentage”, do you mean “lean percentage”? Lean percentage are the common way to describe the content of meat in a carcass, and you are also using the term “lean percentage” later in the manuscript.

Introduction, line 48. Maybe you can define “NK-cells”, if some readers are not familiar with this.

Materials and Methods, 2.1. Animals, lines 93 - 95. You may consider writing the inclusion of the probiotic as g/kg instead og kg/ton. The term “g/kg” sound better in a scientific article. You may also consider writing “2.1. Animals and feed” in line 78.

Materials and Methods, 2.1. Animals, lines 88 – 90. Your description of the probiotic product must be improved. The content of sugar cane molasses seems to be very low. And you do not give any indications of the water content. Why not use “ml/100 g” instead of “ml/g”.

Materials and Methods, 2.2. Carcass and meat quality, lines 102 - 103. In line 79 you write that 120 pigs were used in this study. When you write in lines 102 - 103 that “the same number (60) of carcasses from each group of pigs were selected. …” this is confusing. When you have 129 pigs in total, and two experimental groups, this means 60 pigs per experimental group. You are therefore using all the pigs in the study for the carcass examination, and there is no selection of the pigs to be used. You have to write this in a better way, and also give a better explanation of how you select the pigs for each of the experimental groups as early as in line 79.

Materials and Methods, 2.5. Mineral composition, lines 156 – 158. This sentence must be improved.  

Materials and Methods, 2.6. Microbial determinations, line 180. Your description “Samples of two feeds (for each group)” is unclear and must be improved. Do you mean to say that you took duplicate samples of each of the experimental diets?

Results, 3.3. Carcass and meat quality, line 262. When you write “meat percentage”, do you mean “lean percentage”? See my comments to “line 29”.

Discussion, line 381. Improve the language. Fine a better word than “worse” when you write “Worse quality of meat”.

Discussion, line 393, Improve the language. Do not write “fodder”, “feed” is a better word.

Conclusion, line 444. Writing error, include a space between “influenced” and “carcass”.

Author Response

General comments. Some places in the manuscript you write “EM Bokashi”, and some other places you use “EM®Bokashi”. Please be consequent in how you write this, I prefer “EM®Bokashi”. Has been corrected throughout the publication on “EM®Bokashi”.

Abstract, line 29. When you write “meat percentage”, do you mean “lean percentage”? Lean percentage are the common way to describe the content of meat in a carcass, and you are also using the term “lean percentage” later in the manuscript. Yes. Should be “lean percentage”. We corrected in the text.

 Introduction, line 48. Maybe you can define “NK-cells”, if some readers are not familiar with this. OK, corrected.

 Materials and Methods, 2.1. Animals, lines 93 - 95. You may consider writing the inclusion of the probiotic as g/kg instead og kg/ton. The term “g/kg” sound better in a scientific article. You may also consider writing “2.1. Animals and feed” in line 78. OK. We corrected on “g/kg” and add word „feed” to point 2.1. 

Materials and Methods, 2.1. Animals, lines 88 – 90. Your description of the probiotic product must be improved. The content of sugar cane molasses seems to be very low. And you do not give any indications of the water content. Why not use “ml/100 g” instead of “ml/g”. Please note that the Bokashi probiotic is in a non-liquid, loose form. Maybe that's why the manufacturer does not declare the amount of water in Bokashi. The stated amount of molasses is the manufacturer's declaration on the packaging. As suggested, we changed the unit to ml/100 g.

 Materials and Methods, 2.2. Carcass and meat quality, lines 102 - 103. In line 79 you write that 120 pigs were used in this study. When you write in lines 102 - 103 that “the same number (60) of carcasses from each group of pigs were selected. …” this is confusing. When you have 129 pigs in total, and two experimental groups, this means 60 pigs per experimental group. You are therefore using all the pigs in the study for the carcass examination, and there is no selection of the pigs to be used. You have to write this in a better way, and also give a better explanation of how you select the pigs for each of the experimental groups as early as in line 79. Corrected.

Materials and Methods, 2.5. Mineral composition, lines 156 – 158. This sentence must be improved. Corrected.

 Materials and Methods, 2.6. Microbial determinations, line 180. Your description “Samples of two feeds (for each group)” is unclear and must be improved. Do you mean to say that you took duplicate samples of each of the experimental diets? Yes. Duplicate samples of each two feed mixture for each group of pigs. Corrected.

 Results, 3.3. Carcass and meat quality, line 262. When you write “meat percentage”, do you mean “lean percentage”? See my comments to “line 29”. Correted on “lean percentage”.

 Discussion, line 381. Improve the language. Fine a better word than “worse” when you write “Worse quality of meat”. Corrected

 Discussion, line 393, Improve the language. Do not write “fodder”, “feed” is a better word. Corrected.

 Conclusion, line 444. Writing error, include a space between “influenced” and “carcass”. Corrected.

Round 2

Reviewer 2 Report

The authors did not make the requested changes satisfactorily. No change in the quality of the paper.

Author Response

  1. In the materials and methods, it appears the probiotic was fed to the pigs from d 28 until slaugther. How is this economically viable? considering that the value of better carcass loin with not be cost effective. Please explain why the feeding of the probiotic was done that way. OK. We completed the methodology section about production efficency.
  2. Why didnt you focus on feeding the probiotics in early life of the piglets and then follow them to market to determine the impact. since you already concede that the effects of probiotics are more pronounced in the piglets? see your L42-44. OK, we added a research hypothesis in Introduction.
  3. You explain in L180-184 how the microbiological sampling was done, but you failed to mention at what point in the trial this was done. THis is important because each phase of the trial had a different amount of probiotic per tonne of feed. The material collected for microbiological tests was also related to this probiotic dosage. Accordingly, we have added information in the research methodology.
  4. Considering that the study was done in phases of feeding the probiotics, the statistical analysis section should reflect how this was done, unfortunately that is not the case in the description given in the stats,. please update that section with more details. In our research, we analyzed the effect of the use of the Bokashi probiotic, in the feed technology developed by the breeder, which gives him the best production results, on the final quality of pork, where from the moment of fattening to its end a Bokashi probiotic dose of 3 g / kg of the feed mixture was used. Therefore, the statistically higher use of the probiotic from weaning up to 30 kg of body weight cannot be taken into account for the final quality of pork. However, in the methodology, we made adjustments and additions to the administration of the probiotic during pig growth. The statistical analyzes were better described.
  5. Generally, there was no linkage explaining the mode of action of the probiotics in relation to the impacts on carcass parameters. In other words the paper did not carry a message explaining how probiotics in finisher pigs impacts carcass. In part L. 354-375 we indicated what could have caused the fattening pigs supplemented with probiotics to have higher meatiness and lower fat thickness.
  6. It was not clear in your explanation how the selected probiotics could influence mineral content in the meat in your discussion L407-424. In part L. 413-430 we indicated what could have caused that fattening pigs supplemented with probiotics had a higher, higher concentration of certain micro- and macroelements.
  7. It will be inaccurate the make the conclusions you make no clear inference when the gut microbiota data was taken. This will have influence on the effects on growth performance. We made corrections and additions to the methodology of microbiological tests and conclusions.

Reviewer 4 Report

Comments, review of manuscript id: Animals-1459223-revised.

Title: ”Carcass and pork quality, and gut environment of pigs fed a diet supplemented with the Bokashi probiotic”

The manuscript describes the use of a probiotic supplement in feed for pigs, and its effect on gut health and pork quality. The authors revision of the manuscript is dine according to the comments of the reviewers, but still there may be a small correction in the revised manuscript. See comment below.  

The English language is OK.

Comment.

Discussion, line 388. Here you may have a writing error, I suppose “was cauded” should be “was caused”.

Author Response

OK, corrected

This manuscript is a resubmission of an earlier submission. The following is a list of the peer review reports and author responses from that submission.

Round 1

Reviewer 1 Report

In this manuscript, Rybarczyk and colleagues reported the data from an in vivo study which was conducted to evaluate the effect of feeding so called “effective microorganisms (EMs) supplementation on the slaughter value and mineral content of the muscle of pigs. 

Overall, the study results did not support the conclusion the authors made.  For example, the authors stated that the supplementation of EM bokashi probiotic had significant impact, and positive impact on pig gut microbiota.  Without having 16S rRNA gene sequencing data on characterizing the fecal or gut microbiota, I did not understand how the authors made such a conclusion base on just conventional culture technique. 

Since the EM bokashi probiotic was supplemented in the diet of the pigs, it is important to first show how the probiotic supplementation impacted the gut microbiota. This can be done by sampling feces before and several time points after the probiotic supplementation. Without such data, it is challenging to explain the difference in the carcass and meat quality between control and EM bokashi probiotic supplemented groups. 

No detailed information was provided with respect to a specific time of feed sampling. 

The data presented in this manuscript is not sufficient enough to be a full paper.

The manuscript is not well written with many language errors, which made it difficult to understand.  For example.

  • Line 13-17: the sentence is too long and it doesn’t read well.
  • Line 18-20: this sentence is not correct.
  • Many more……

The introduction section is poorly written with no hypothesis.

The details on the experimental design are not provided. 

  • Lines74-81: a total of 120 commercial hybrid pigs ….. the animal were allocated to 20 pens……12 animals in each replication….. so how many animals were actually in each treatment and control group?

Many confusing and undefined terminology are used in the manuscript.

  • What is effective microorganism?
  • What is technological yield?

Why lactic acid bacteria were cultured on Trypticase Soya broth? Why not Lactobacilli MRS Broth? Why the incubation temperature was set at 30 C not 37 C 

Reviewer 2 Report

The article is interesting. The use of the EM Bokashi probiotic improved the quality of carcasses and the colour of the meat, but reduced its technological usefulness. Since the applied EM Bokashi increased drip loss, shore force, and lowered the protein content, the question is whether the benefits of its use (effects on the gastrointestinal tract and immune system) outweigh the disadvantages.

The article lacked the general composition of compound feed. The Authors only state that the feed did not contain any antibiotics for therapeutic purposes. 

No indication of the age of pigs on the day of slaughter. Was it an average of 164 days or more?

Since the studies have a practical aspect, it would be good to give at least in one sentence the results obtained in fattening: ADG and feed consumption for pigs from both groups.

Did there be diarrhoea in pigs during fattening?

I believe that the determination of control group C and experimental group E (especially in Tables 2-4) will be more transparent.

L 189 is "22oC" should be "22oC"

L 461 is "2016" should be "2016"

L 475 is " Rocz. Nauk. Pol. Towarz. Zootech." should be "Rocz. Naukowe PTZ"